# Liver Cell Type-Specific Targeting by Nanoformulations for Therapeutic Applications

**DOI:** 10.3390/ijms241411869

**Published:** 2023-07-24

**Authors:** Leonard Kaps, María José Limeres, Paul Schneider, Malin Svensson, Yanira Zeyn, Silvia Fraude, Maximiliano L. Cacicedo, Peter R. Galle, Stephan Gehring, Matthias Bros

**Affiliations:** 1I. Department of Medicine, University Medical Center Mainz, Langenbeckstrasse 1, 55131 Mainz, Germany; leonardkaps@googlemail.com (L.K.); paschnei@students.uni-mainz.de (P.S.); galle@uni-mainz.de (P.R.G.); 2Children’s Hospital, University Medical Center, Langenbeckstrasse 1, 55131 Mainz, Germany; mj.limeres@uni-mainz.de (M.J.L.); malin.svensson@uni-mainz.de (M.S.); silvia.fraude@unimedizin-mainz.de (S.F.); mcaciced@uni-mainz.de (M.L.C.); stephan.gehring@uni-mainz.de (S.G.); 3Department of Dermatology, University Medical Center Mainz, Langenbeckstrasse 1, 55131 Mainz, Germany; yanira.zeyn@uni-mainz.de

**Keywords:** liver, non-parenchymal cells, Kupffer cell, liver sinusoidal endothelial cell, nanoformulation, hepatitis, fibrosis, hepatocellular carcinoma, liver metastasis

## Abstract

Hepatocytes exert pivotal roles in metabolism, protein synthesis and detoxification. Non-parenchymal liver cells (NPCs), largely comprising macrophages, dendritic cells, hepatic stellate cells and liver sinusoidal cells (LSECs), serve to induce immunological tolerance. Therefore, the liver is an important target for therapeutic approaches, in case of both (inflammatory) metabolic diseases and immunological disorders. This review aims to summarize current preclinical nanodrug-based approaches for the treatment of liver disorders. So far, nano-vaccines that aim to induce hepatitis virus-specific immune responses and nanoformulated adjuvants to overcome the default tolerogenic state of liver NPCs for the treatment of chronic hepatitis have been tested. Moreover, liver cancer may be treated using nanodrugs which specifically target and kill tumor cells. Alternatively, nanodrugs may target and reprogram or deplete immunosuppressive cells of the tumor microenvironment, such as tumor-associated macrophages. Here, combination therapies have been demonstrated to yield synergistic effects. In the case of autoimmune hepatitis and other inflammatory liver diseases, anti-inflammatory agents can be encapsulated into nanoparticles to dampen inflammatory processes specifically in the liver. Finally, the tolerance-promoting activity especially of LSECs has been exploited to induce antigen-specific tolerance for the treatment of allergic and autoimmune diseases.

## 1. Introduction

The liver constitutes one of the largest organs of the body and exerts numerous essential functions associated with digestion, metabolism, protein synthesis and the detoxification of xenobiotics [1]. Hence, the liver is a primary site of various metabolic disorders [2]. Further, chronic hepatitis [3] and metabolic disorders [4] may result in liver fibrosis or even cirrhosis. Cirrhosis is considered a premalignant state as 80% of hepatocellular carcinoma (HCC), the most frequent type of primary liver cancer, develop in cirrhotic livers [5]. In addition, due to anatomical reasons, the liver is also a primary organ of tumor metastasis [6]. The therapy of liver tumors is complicated by the default tolerogenic state of liver-resident non-parenchymal cells (NPCs) [7]. Hence, the liver, with its different specialized cell types, is a highly interesting target organ for cell-type-addressing nanoformulations, thereby avoiding adverse side effects, such as systemic toxicity [8]. However, as outlined in this review, the structure of the liver has proven a challenge for nano-based cell-targeting approaches.

On the one hand, the liver acts as a biological barrier and may sequestrate the vast majority of administered nanoparticles (NPs) from the bloodstream, when particles (>6 nm) are not eliminated by the kidney [9]. However, on the other hand, within the liver NPs need to pass several biological barriers hindering their successful uptake by the different liver cell populations [10]. In this regard, NPs intended to address hepatocytes need to cross the sinusoidal fenestrations of liver sinusoidal endothelial cells (LSECs) [11,12]. Furthermore, Kupffer cells (KCs), that constitute the major liver-resident macrophage population [13], and LSECs express various types of Fc and scavenger receptors [14] for the efficient uptake of larger KC [13] and smaller LSEC [15] macromolecules, and consequently also NPs [9]. Therefore, hepatocyte-targeting NPs need to circumvent unwanted uptake by the aforementioned liver NPC populations.

This review aims to summarize the strategies currently evaluated in preclinical in vivo models for the treatment of liver diseases with nanoformulations that may allow the poor solubility of drugs to be overcome by encapsulation and enable the codelivery of distinct compounds that act on different signaling pathways or effector molecules into the same cell to yield synergistic effects [16,17]. Consequently, nanoformulations are also intended to prevent systemic toxicity, as frequently observed, e.g., for systemically applied anti-cancer drugs [18,19]. Therefore, nanoformulations need to address their cellular target either via passive or active targeting [20]. Whereas the former is determined by the intrinsic properties of the NP-like size and charge, active targeting is achieved by the attachment of receptor-binding moieties on the NP surface. For the treatment of liver-associated disorders, nanoformulations need to exert disease-specific biological effects; for instance, in the case of virus-induced hepatitis nano-vaccines aim to evoke antiviral immune responses [21,22], whereas in the case of autoimmune hepatitis nanodrugs are intended to dampen inflammation [23].

Liver fibrosis, i.e., the accumulation of extracellular matrix (ECM) proteins due to the activation of hepatic stellate cells (HSCs), is commonly observed in the course of chronic liver diseases, such as non-alcoholic fatty liver disease [24]. Extensive liver fibrosis may result in cirrhosis, which, in turn, predisposes the liver to HCC [25]. Hence, a number of nanoformulations that revert HSC activation are under development. Further, in the case of liver cancer, nanodrugs may either directly address tumor cells to yield tumor cell-restricted cytotoxicity [26], or aim to reprogram tumor-associated cells that support tumor immune evasion [27]. In the case of liver anti-tumor therapy, the efficacy of nanoformulations has been assayed in several cases in combination with an immune checkpoint blockade (ICB) that aims to enhance T cell responses to improve therapeutic effects [28].

Finally, we will also discuss how the common default tolerance-inducing role of the various liver NPC populations (KCs [29]; LSECs [30]; dendritic cells (DCs) [31]) has been exploited to induce antigen-specific regulatory T cells (Tregs) for the treatment of allergies [32] and autoimmune diseases [33].

## 2. Nanodrugs for the Treatment of Liver Disease

### 2.1. Viral Hepatitis

Hepatitis constitutes a state of liver inflammation that is induced by a variety of noxae, such as viral infections, alcohol abuse and autoimmune reactions [34]. The pathogenesis of liver damage in hepatitis B and autoimmune hepatitis, as the most frequent kinds of hepatitis, shows certain similarities [35]. In both diseases, cells of the immune system cause inflammatory reactions that damage the liver. In chronic hepatitis B, virus antigen-specific T cells are considered to be the main driver [36]. These virus-specific lymphocytes are not capable of eradicating the virus but recognize virus-specific proteins in the liver which cause hepatocyte damage [37]. Under these inflammatory conditions, KCs promote activation the of HSCs, which transform into myofibroblasts that produce large amounts of ECM proteins, which replace the parenchymal tissue [38]. However, a plethora of other NPCs (e.g., liver-resident macrophages) contribute to fibrogenesis in the liver. A comprehensive review of the underlying mechanisms and involved cell types was recently published [39]. When fibrogenesis persists, liver cirrhosis, as an end-stage of virtually every chronic liver disease, can occur [40]. Portal hypertension is a consequence of cirrhosis and is responsible for the most severe clinical complications, such as bleeding from gastro-esophageal varices and hepatic encephalopathy [41].

Viral hepatitis can be caused by five different hepatitis viruses, which have different transmission routes and variable clinical courses [42]. The pathogenicity of hepatitis G for humans is unclear and is therefore not further discussed. All of them are single-stranded RNA viruses, except hepatitis B (HBV), which contains its genetic material as stable DNA [43]. Hepatitis D virus (HDV) is a satellite virus, as it can propagate only in the presence of HBV because it requires HBV surface antigens in its viral envelope in order to replicate [44]. All types of hepatitis viruses can lead to fulminant hepatitis, while only HBV, hepatitis C virus (HCV) and, in rare cases, hepatitis E virus (HEV) can cause chronic infections, constituting a risk of cirrhosis and HCC [45,46]. Co-infection of HBV and HDV is considered the most serious type of viral hepatitis, which leads to a complicated clinical course with fast progression to cirrhosis [47].

Hepatitis A virus (HAV) and HEV are transmitted by the fecal–oral route. HBV infections occur as a result of exposure to contaminated blood and body fluids [48]. In practice, HBV is mostly transmitted vertically (mother-to-child). HCV is transmitted by contaminated blood (e.g., needle sharing, contaminated blood transfusion) or, in rare cases, genital secretions (mostly reported for men who have sex with men) [49].

The clinical course of the infection depends on the type of hepatitis virus. HAV and HEV infections play a minor role in clinics because most patients spontaneously recover, and only at-risk groups may develop acute liver failure [50]. The treatment of hepatitis C dramatically improved in 2014 when effective antiviral regimes were clinically approved [51]. Oral tablet regimes achieve very high curing rates and novel drug loaded β-cyclodextrin NPs may even improve the safety and tolerability of treatments [52]. Accordingly, the World Health Organization is confident that hepatitis C may one day be eradicated [53]. In contrast, there is a risk of hepatitis B reactivation even in cured patients [54]. One of the main reasons for the refractory nature of hepatitis B towards treatment is the covalently closed circular DNA (ccc DNA) of HBV, which serves as a template for all transcripts and persists in the nucleus of hepatocytes [55]. The reactivation of HBV can occur under immunosuppression and is a potentially life-threatening condition when liver failure occurs [56].

Since HBV cannot be completely eliminated by the immune system, prophylactic vaccination is a pillar of all prevention strategies [57]. One strategy to achieve effective protection is vaccines that include adjuvants which induce a strong immune response. [58]. Traditional hepatitis B vaccines use aluminum compounds as an adjuvant to induce a robust humoral response [59]. However, vaccines with this kind of adjuvant cannot yield an adequate cellular immune response to effectively recognize and eliminate HBV-infected cells [60]. Therefore, novel adjuvants which increase not only humoral but also cellular immune responses are needed. For example, Pu Shan and coworkers developed a saponin-based nano-adjuvant which induced stronger humoral and cellular immune responses than aluminum-based adjuvants in a murine hepatitis model [61]. Further, Qiao et al. generated nano-vaccines composed of chitosan plus heparin that encapsulated the HBV surface and core antigen, respectively, and contained immunostimulatory CpG oligo in addition [62]. In a mouse model of chronic hepatitis B infection these two types of nano-vaccines were coapplied. After subcutaneous injection, the two nanovaccines targeted the draining lymph nodes, achieving the seroclearance of HBV surface antigens in most mice. In addition, vaccination induced long-term immune memory and protected the mice from HBV reinfection. Interferon-α (IFN-α), either in its soluble form or conjugated to polyethylene glycol (PEG), is used to treat chronic hepatitis B but only 20–40% of patients respond well, and the treatment is associated with side effects due to its systemic administration [63]. Fayes and coworkers demonstrated the liver-restricted induction of IFN-α by the toll-like receptor 7 (ligand imiquimod (IMQ)) when delivered as a nanoformulation [64]. To this end, IMQ was encapsulated into liposomal vesicles, which conferred hepatocyte-specific uptake, and into the anionic liposomes that mediated KC-restricted internalization. In a mouse model of chronic hepatitis B, both types of NPs yielded liver-specific IFN-α production and concomitantly reduced HBV DNA serum levels.

### 2.2. Autoimmune Hepatitis

The exact etiology of autoimmune hepatitis is not known [65]. A genetic predisposition in combination with environmental factors, viral infection and drugs may trigger this disorder. Hepatitis is considered to be of autoimmune origin only when viral hepatitis can be excluded, reflecting the high degree of histological and clinical similarities between viral and autoimmune hepatitis [35]. In autoimmune hepatitis, elevated titers of autoantibodies (e.g., antinuclear and anti-smooth muscle antibodies) and increased immunoglobulin G levels are observed, which are normally not present in viral hepatitis and, thus, can be used in differential diagnosis [66].

The mainstay of treatment is immunosuppressive drugs, such as glucocorticoids, azathioprine or methotrexate and mycophenolate in severe cases [67]. However, the systemic use of immunosuppressive drugs provokes the risk of side effects, such as Cushing’s syndrome in the case of glucocorticoids [68], or opportunistic infections when applying azathioprine, methotrexate and mycophenolate, respectively [69]. An interesting concept to circumvent the systemic effects of glucocorticoids is dexamethasone-loaded NPs [23], which display an enhanced liver tropism when composed of avidin and nucleic acid [70,71]. Upon intraperitoneal administration, these NPs primarily addressed the liver and high levels of dexamethasone were detected in livers but not in sera [70]. Another interesting application of immunosuppressive NPs is to alleviate the side effects of ICB in cancer therapy [72]. Poly(L-lactic-co-glycolic acid) (PLGA) NPs coated with the anti-programmed cell death protein (PD-)1 antibody did not induce the hepatotoxicity that may occur in the course of ICB therapy [73] without affecting the antitumor effect of ICB in a murine tumor model [72].

### 2.3. Non-Alcoholic Fatty Liver Disease (NAFLD)

With an estimated global prevalence of almost ~25%, non-alcoholic fatty liver disease (NAFLD) is a leading cause of liver disease worldwide [74]. NAFLD ranges from steatosis without or with mild inflammation [75] to advanced stage non-alcoholic steatohepatitis (NASH), which is characterized by the necroinflammation and ballooning of hepatocytes [76]. NASH can progress to fibrosis and, ultimately, cirrhosis with the risk of HCC or intrahepatic cholangiocellular carcinoma and death [77]. The risk factors for NAFLD are obesity, type 2 diabetes mellitus and metabolic syndrome.

Despite great research efforts, no drug has been approved yet for NAFLD or NASH treatment. However, several clinical studies are ongoing to evaluate potential drug candidates. For instance, the anti-diabetic drug pioglitazone and vitamin E (i.e., δ-tocotrienol and α-tocopherol) demonstrated a therapeutic effect (e.g., an improvement in hepatic steatosis) in patients with NAFLD. Obeticholic acid, a farnesoid X nuclear receptor, improved the histological features of NASH but long-term results are still pending [78]. However, weight loss and dietary modifications are currently the therapeutic mainstay for these liver disorders [79]. Plant-derived celastrol (CEL) is a promising anti-inflammatory and anti-obesity drug but its low oral bioavailability hampers its clinical use [80]. Albumin-based NPs were synthesized to facilitate the liver-specific delivery of CEL [81]. To this end, CEL was encapsulated into lactosylated bovine serum albumin using high pressure homogenization. CEL-NP showed improved uptake into hepatocytes and enhanced hepatic deposition compared to free CEL. Further, encapsulated CEL outperformed free CEL in reducing lipid deposition, ameliorating liver function and enhancing insulin sensitivity in a murine model of diet-induced NAFLD without causing side effects. Besides, the genes for lipogenesis and lipid transport were upregulated by treatment.

Reducing calorie uptake in overweight patients may be an additional application of NPs. α-glucosidase is an approved drug for limiting the absorption of polysaccharides and disaccharides but it is ineffective for monosaccharides [82]. However, the boronic acid-containing polymer nanocomplex (Nano-Poly-BA) absorbed all types of saccharides and, thus, could be used to avoid their intestinal uptake [83]. Nano-Poly-BA showed remarkable after-meal blood glucose reductions in type 1 and type 2 diabetic mouse models when mice were fed with coke, blueberry jam or porridge. Orally administered Nano-Poly-BA proved to be non-absorbable and non-toxic. The authors concluded that Nano-Poly-BA may help diabetic, overweight and even healthy people to manage their sugar intake. In the methionine-choline-deficient diet mouse model the administration of a liposomal formulation equipped with a sphingosine receptor-targeting moiety for HSC targeting [84] and containing short chain C6-ceramide normalized dysregulated lipid homeostasis [85].

In the course of fibrosis-inducing events, activated LSECs close fenestration gaps and are no longer able to keep HSCs in a quiescent state; they therefore transdifferentiate to myofibroblasts and produce ECM proteins to a large extent [86]. HSCs store large amounts of vitamin A in the liver [87]. In line, several studies demonstrated the vitamin A-mediated targeting of various types of NPs to HSCs, which inhibited HSC activation-dependent fibrosis in various rodent models (Figure 1). For example, Perri et al. employed silver NPs that contained a nitric oxide donor [88] to trigger soluble guanylyl cyclase signaling [89]. Liposomes encapsulating sterol regulatory element-binding protein 2-specific small interfering (si)RNA and anti-miR-33a [90] prevented TGF-β-mediated HSC activation [91]. Qiao and coworkers [92] developed core-shell polymeric micelles that codelivered the antioxidant silibinin [93] and collagen1a1-specific siRNA [94], thereby attenuating ECM formation.

As an alternative approach, Zhang et al. applied liposomes with hyaluronic acid, previously reported to target LSECs and HSCs [95] (Table 1), and containing simvastatin to revert LSEC capillarization [96]. Subsequently, a second liposomal nano-carrier was employed to deliver collagen 1-specific siRNA to HSCs and, thereby, attenuate ECM formation.

In addition, untargeted nanohydrogel particles, which efficiently accumulate in the liver, hold promise for antifibrotic treatment. Two intravenous injections of anti-collagen 1-specific siRNA-loaded nanohydrogel particles (2 mg/kg siRNA) reduced the hepatic collagen load to the levels of healthy control mice [97,98].

**Figure 1 ijms-24-11869-f001:**
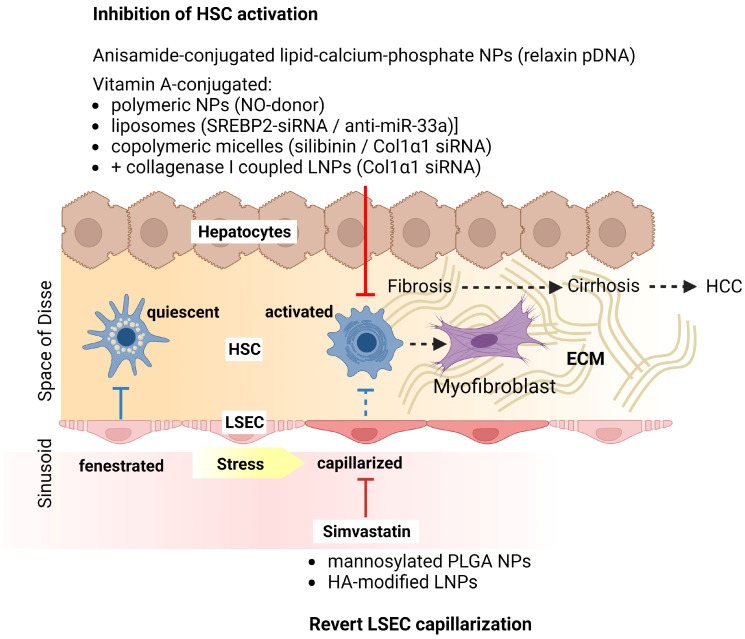
Nanoformulations for the inhibition of liver cirrhosis. LSEC serve to keep HSC in a quiescent state (blocking arrow). Various stress factors result in the loss of LSEC fenestration, termed capillarization, and the loss of LSEC-mediated HSC inactivation (dashed blocking arrow). Activated HSCs transdifferentiate to myofibroblasts and generate ECM proteins, resulting in fibrosis, cirrhosis and, ultimately, HCC. Fibrosis-counteracting nanoformulations target LSECs, e.g., via mannose [99] and hyaluronic acid [96], and deliver simvastatin to revert capillarization and to reestablish HSC-inhibition by LSECs. Alternatively, nanodrugs may target HSCs especially via vitamin A, and inhibit ECM production, e.g., by the delivery of drugs that inhibit collagen expression at various levels [88,90,92,96,100]. ECM, extracellular matrix; HA, hyaluronic acid; HCC, hepatocellular carcinoma; HSC, hepatic stellate cell; LNP, lipid nanoparticle; LSEC, liver sinusoidal endothelial cell; miR, micro RNA; NP, nanoparticle; PLGA, poly(L-lactic-co-glycolic acid); siRNA, small interfering RNA. Created with BioRender.com (4 May 2023).

**Table 1 ijms-24-11869-t001:** Target receptors and receptor-binding moieties for liver cell type-specific addressing by NPs.

Cell Type	Target Receptor	Targeting Moiety	Ref.
**HCC**	α_v_β_3_ integrin receptor [101]	Cyclic RGD peptide [102]	[103,104]
	Asialoglycoprotein receptor [105]	N-acetylgalactosamine	[106,107,108]
	CD44 [109]	Hyaluronic acid	[110,111,112]
	CXC motif chemokine receptor 4 (CXCR4) [113]	CTCE-9908 peptide [102]	[114]
	CD147 [115]	Anti-CD147 antibody	[116,117,118]
	EGFR [119]	GE11 peptide [120]	[121]
	Folate receptor [122]	Folate	[123,124,125]
	Glycyrrhetinic acid receptor [126]	Glycyrrhizic acid	[9,127,128,129]
	Glypican3 (GPC3) [130]	Membrane of GPC3-specific CAR-T cells	[131]
		Anti-GPC3 antibody	[132]
	Nucleolin receptor [133]	AS1411 aptamer [134]	[135]
	Transferin receptor [136],	Transferin,	[137]
	Unknown	TLS11a aptamer [138], SP94 peptide [139],	[140,141]
**HSC**	Mannose-6-phosphate receptor [142]	Mannose-6-phosphate	[143,144,145]
	Retinol-binding protein receptor [146,147]	Retinol-binding protein	[148,149]
	Collagen type VI receptor [150]	Cyclic arginine-glycine-aspartate peptide	[150,151]
	CD44	Hyaluronic acid	[95,129,152]
	Platelet-derived growth factor beta receptor [153]	pPB peptide [154]	[155,156,157]
**KC**	CD163 [158]	Anti-CD163 antibody	[159,160]
	CD206 [158]	Mannose	[137,161,162,163]
**LSEC**	CD44	Hyaluronic acid	[95]
	CD206 [164]	Mannose	[99,165,166]
	Stabilin-1, -2 [167,168]	Cholesteryl oleate, ApoB peptide	[32,165,169,170]

### 2.4. Hepatic Tumors/Metastasis

Liver cancer has been ranked as the third leading cause of cancer deaths worldwide and its incidence is further increasing [5]. HCC is the most prevalent primary liver cancer in adults, which accounts for up to 90% of cases, followed by cholangiocarcinoma (CCC) [171]. Only the minority are hepatocellular cholangiocarcinoma (a mixed form of HCC and CCC), angiosarcoma and hepatoblastoma. HCC originates predominantly from hepatocytes and transformed activated hepatic progenitor cells, while CCC is a bile duct neoplasm [172,173]. The predominant risk factor for HCC is parenchymal damage. Patients with cirrhosis have a high annual risk of ~1–6% of developing HCC [174]. Besides cirrhosis, an important risk factor for CCC is autoimmune disease of the bile duct system (e.g., primary biliary cholangitis or primary sclerosing cholangitis) [175,176]. The mainstay of treatment is a combination of the immune checkpoint programmed cell death-1 ligand (PD-L1)-blocking antibody, atezolizumab [177], with the anti-angiogenic vascular endothelial growth factor (VEGF)-A-neutralizing antibody, bevacizumab [178], in patients with unresectable HCC, reaching an overall survival of 13.2 months [179]. For unresectable CCC, chemotherapy combined with the PD-L1 immune checkpoint inhibitor, durvalumab [177], could improve overall survival, progression-free survival and objective response rates compared to chemotherapy alone, while the estimated 24-month survival rates remained low (24%) [180]. Besides primary cancer, the liver is susceptible to hematogenous metastasis due to its high blood perfusion [6]. Hence, numerous cancer patients present with liver metastasis in the course of disease progression, for example, colorectal cancer patients show liver metastasis in up to half of all cases [181]. LSECs play an important role in liver metastasis by expressing adhesion receptors, such as E-selectin, thereby engaging extravasating tumor cells [182,183].

Liver NPCs and infiltrating immunoregulatory cells promote the growth of HCC and liver metastasis by the release of mitogenic factors, such as hepatocyte growth factor (HGF) [184] and proangiogenic factors, such as VEGF [185]. In addition, anti-inflammatory cytokines, such as interleukin (IL)-10 [186] and transforming growth factor (TGF)-β [187] contribute to establish an immunosuppressive tumor microenvironment (TME) [188,189] that consists on cellular level of immune cells (see below), endothelial cells, HSCs and cancer-associated fibroblasts (CAFs) [190]. Interestingly, CAFs originate from diverse cell types, including mesenchymal stem cells, fibroblasts, endothelial and epithelial cells, as well as hepatocytes that (trans)differentiate under the influence of TME-derived cues [191], but the predominant cellular source is activated HSCs [192]. CAFs support tumor growth through the release of various factors, such as TGF-β [193], HGF [194] and IL-6 [195], but also chemokines, such as CC-chemokine ligand (CCL)2 [196], which attracts monocytes that may polarize towards tumor-associated macrophages (TAMs) [197]. Chemokines also attract myeloid-derived suppressor cells (MDSCs) [198] that arise in the bone marrow under the influence of TME-derived mediators, such as IL-1β and IL-6 from granulocytic and monocytic progenitors, respectively [199]. Polymorphonuclear neutrophilic leukocyte (PMN) may acquire an immunosuppressive state under the influence of the TME as well, similar to PMN-like MDSCs [200]. Besides immunoregulatory cells of myeloid origin, Treg cells are also attracted towards the tumor site, e.g., by CCL22 [201].

So far, the vast majority of preclinically evaluated nanoformulations for the treatment of liver cancer have focused on the delivery of cytotoxic agents into the tumor (Figure 2). In most cases these nanoformulations passively target the liver as demonstrated, for example, by gold NPs [202], micelles composed of diblock copolymers of methoxy poly (ethylene glycol)-poly(L-lactic acid) (mPEG-PLA) [203] and β-tricalcium phosphate NPs [204].

However, a growing number of studies have been aimed at targeting active tumor cells by decorating nanocarriers with moieties that address receptors highly expressed by tumor cells [205], such as, for example RGD peptides [103] that engage integrin αvβ3 [206], epidermal growth factor receptor binding peptides [121], the folate receptor [123] and cluster of differentiation (CD)44- [110] and CD147- [207] specific antibody fragments [116]. In the case of the latter study, additional decoration with a cell-penetrating peptide [208] improved cellular uptake [116]. The HCC receptors and the concordant ligands used to achieve HCC-specific NP delivery are listed in Table 1. As an alternative approach for HCC targeting, in screening studies, aptamers with an intrinsic HCC-binding affinity to as-yet undefined receptors have been identified and used for HCC targeting by NP [138,139].

**Figure 2 ijms-24-11869-f002:**
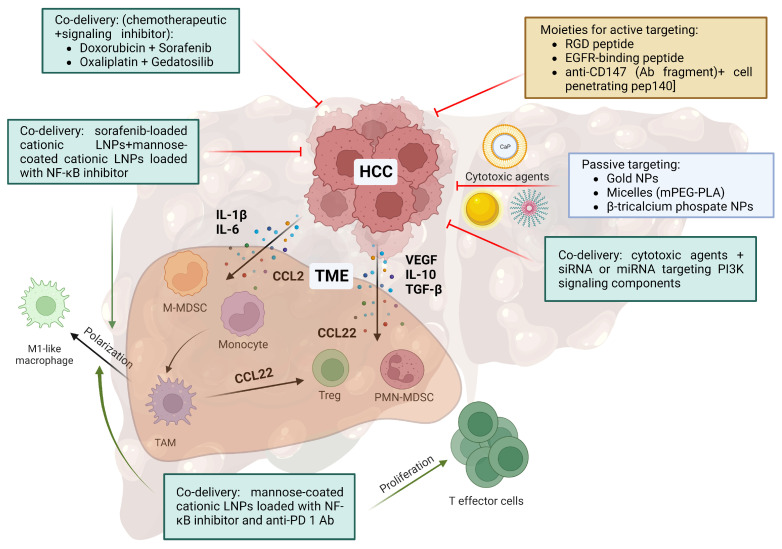
Nanodrugs for liver tumor therapy. Tumors establish an immunosuppressive tumor microenvironment (TME) by attracting immune cells that are reprogrammed to exert immunoinhibitory effects. Nanoformulations may either passively ([202,203,204]) or actively ([103,116,121,162]) target tumor cells to deliver cytotoxic drugs alone ([106,162,209]) or in combination with signaling inhibiting agents ([210,211]). As an alternative, nanoformulations may target components of the TME and cause their differentiation to acquire anti-tumor activity ([162,212]). Ab, antibody; CCL, CC-chemokine ligand; EGFR, epidermal growth factor receptor; IL, interleukin; LNP, lipid nanoparticle; mMDSCs, monocytic myeloid-derived suppressor cells; miRNA, micro RNA; NF-κB, nuclear factor kappa-light-chain-enhancer of activated B cells; NP, nanoparticle; PD-1, programmed cell death protein 1; PEG, polyethylene glycol; PLA, poly(L-lactic acid); PMN, polymorphonuclear neutrophilic leukocyte; siRNA, small interfering RNA; TAM, tumor-associated macrophage; VEGF, vascular endothelial growth factor. Created with BioRender.com (4 May 2023).

In most cases, tumor-targeting nanoformulations aimed to improve the tumor-specific effects of chemotherapeutics by attenuating systemic toxicity [213]. However, in a growing number of studies the potential of nanocarriers to codeliver different agents with distinct molecular targets to achieve synergistic effects has been demonstrated. In many cases, a chemotherapeutic drug (e.g., doxorubicin or oxaliplatin) is codelivered with a signal transduction inhibitor (e.g., sorafenib) that inhibits the pathways upregulated in cancer [214], such as Raf kinases [106] or phosphoinositide 3-kinases (PI3Ks)/mechanistic targets of rapamycin (mTORs) [209]. In further combination approaches, cytotoxic agents were delivered with metal-containing nano-carriers to enable chemo-photothermal therapy [215,216]. As a rather novel approach, cytotoxic drugs have been codelivered with small interfering (si)RNA to inhibit, e.g., VEGF-mediated neoangiogenesis [210], and with micro (mi)RNA to inhibit the components of the PI3K signaling pathway [211].

Furthermore, the TME has been targeted by nanoformulations with the aim of decreasing its immunosuppressive character. Hypoxia is a hallmark of larger solid tumors which promotes the immunosuppressive state of the TME, both by reprograming, e.g., macrophages towards TAMs [217] and by directly inhibiting the activity of tumor-infiltrating leukocytes, e.g., via metabolic cues [218]. Furthermore, hypoxia drives the expression of proangiogenic genes, such as VEGFs, resulting in the neovascularization of the tumor [219]. Chang et al. generated NPs that consisted of an MnO_2_ core to generate oxygen from hydrogen peroxide decomposition and a lipid–PLGA shell containing sorafenib as an angiogenesis inhibitor [220]. HCC targeting was achieved by conjugating the HCC-targeting SP94 peptide [139]. This composite nanoformulation inhibited tumor growth, which was associated with the repolarization of TAMs towards M1-like macrophages, and elevated the tumor infiltration of CD8^+^ cytotoxic T lymphocytes (CTLs), as well as impairing tumor vascularization. Wang and coworkers observed that so-called high-density lipoproteins displayed intrinsic HCC- and macrophage-targeting properties [221]. In the same study, the codelivery of vadimezan to disrupt newly formed blood vessels within the tumor [222] and of the cytotoxic drug, gemcitabine [223], caused a release of danger-associated molecules which, in turn, also resulted in a shift of TAMs towards M1-like macrophages and increased CTL infiltration.

In a TAM-focused approach, Wang and coworkers [162] demonstrated the synergistic anti-tumor effects of sorafenib-loaded cationic lipid (L)NPs for direct HCC killing and coapplied cationic LNPs coated with mannose for TAM targeting that were loaded with the nuclear factor kappa-light-chain-enhancer of activated B-cells (NF-κB) inhibitor, IMD-0354 [224], to achieve TAM reprograming towards M1-like macrophages exerting anti-tumor activity [225]. Likewise, NPs composed of cationic chitosan and anionic poly-glutamic acid and loaded with doxorubicin for tumor cell killing and IL-12 to reprogram TAMs towards M1 macrophages were demonstrated to accumulate in the liver and to exert therapeutic efficacy in an HCC model [226]. In the case of HCCs, TAMs are the major PD-L1-expressing cell type within the TME, and limit T cell activity by the engagement of PD-1 [227]. PD-L1-binding immune checkpoint inhibitors were reported to reduce the number of TAMs and to improve tumor infiltration by CD4^+^ T helper cells (Th) and CTLs [228]. As an alternative to ICB treatment, which has been associated with adverse side effects [229], LNPs were loaded with a plasmid DNA that encoded a solubilized PD-1 receptor to bind to PD-L1 [230]. These NPs accumulated in the liver and exerted therapeutic activity in HCC models, which was associated with decreased TAM numbers and the attenuated infiltration of PMN-related MDSCs but enhanced CTL infiltration. In a combination approach, LNPs coated with mannose for TAM targeting were co-loaded with a NF-κB inhibitor to reprogram TAMs and with a PD-1-blocking antibody to enhance T effector cell proliferation [212] into nanogels that released their content in a matrix metalloproteinase 2-responsive manner at the tumor site [163]. Further strategies to reprogram and deplete TAMs in HCC have recently been reviewed [231].

In HCC both the loss of LSEC fenestration and the accumulating ECM may prevent drug delivery to the tumor site. Simvastatin has been demonstrated to restore LSEC capillarization by activating the transcription factor Kruppel-like factor (KLF)2 [232]. KLF2, in turn, activated endothelial nitric oxide synthase and the derived NO triggered soluble guanylyl cyclase-dependent signaling [233]. Yu and coworkers demonstrated in a murine HCC model that PLGA-based NPs loaded with simvastatin and coated with mannose to target LSECs reverted LSEC capillarization and deactivated HSCs [99]. Moreover, simvastatin induced the production of the chemokine CXCL16 by LSECs, which attracted natural killer T cells that exerted anti-tumor activity. Lipid calcium-phosphate NPs, which were conjugated with aminoethyl anisamide suggested to engage the sigma-1 receptor found to be highly expressed by activated HSCs [234], delivered plasmid DNA encoding the anti-fibrotic peptide, relaxin [100], thereby reverting fibrosis within the tumor lesion [235]. Synergistic anti-tumor effects were achieved when combining this approach with a PD-L1 blockade. Further studies on nanodrug delivery to HSCs and the relevance of these strategies to address closely related CAFs are discussed by Kaps and Schuppan [236].

### 2.5. Tolerance Induction

Tolerance towards self-antigens is conferred largely in the thymus, resulting in the deletion of T cells whose antigen-recognizing T cell receptor displays a high affinity towards a self-protein-derived peptide antigen, and, in case of intermediate affinity, promotes their differentiation towards Tregs [237]. In the periphery, T cell tolerance towards self-proteins, as well as harmless environmental antigens, is induced and maintained by immature and tolerogenic DCs that anergize reactive T cells or convert them into Tregs [238]. The loss of tolerance towards self-antigens results in autoimmune diseases [239] and, in the case of environmental antigens, in allergies [240]. The only established therapeutic treatment option for allergies is based on the subcutaneous or sublingual delivery of the allergen to (immature) cutaneous DCs to reestablish tolerance [241]. However, this approach requires several years of treatment and is not always successful. For autoimmune diseases no established therapy is available yet [242]. Ongoing clinical trials aim to reestablish self-tolerance by tolerizing in vitro monocyte-derived DCs, e.g., with anti-inflammatory cytokines, such as IL-10, or drugs, such as glucocorticoids, followed by the reinfusion of these DCs into the patient [243,244]. In many cases, these DCs are loaded beforehand with relevant autoantigens. Preclinical trials aim to address DCs in vivo by employing nano-vaccines that deliver an auto-antigen or allergen, respectively, and in many cases also contain a tolerance-promoting agent [245].

More recently, several studies have aimed to exploit the intrinsic tolerance-promoting activity of liver NPCs [246] as an alternative to DC tolerization, since the former are known to induce Tregs [247] and to inhibit T effector cells [248] in an antigen-specific manner. Through comparative biodistribution experiments, Carambio and coworkers identified the ability of iron oxide-based NPs coated with poly(maleic acid-alt-1-octadecene) [249] to passively target LSECs in vivo after systemic administration [250]. In a mouse model of multiple sclerosis, experimental autoimmune encephalomyelitis (EAE) was conducted. These NPs, when coated with the relevant autoantigen used for EAE induction, inhibited the onset of disease when applied one day after EAE induction in myelin basic protein-immunized B10.PL mice and delayed the onset of EAE in myelin oligodendrocyte glycoprotein-immunized C7BL/6 mice. Inhibitory effects in C7BL/6 mice were also apparent in a therapeutic setting. In either case, LSEC-induced CD4^+^ Tregs played a major role in EAE inhibition. In a subsequent study, Carambio et al. demonstrated the efficacy of this tolerization approach in a model of CD8^+^ T cell-driven autoimmune cholangitis by inhibiting CD8^+^ T cell cytotoxicity and inflammatory cytokine production [33].

Concerning allergy models, Xu and coworkers generated LNPs [166] that were decorated with mannose to predominantly address LSECs [166], which display high mannose receptor density [14,251]. The administration of mannosylated LNPs loaded with peanut allergen (Ara h2) peptide-encoding mRNA resulted in the generation of IL-10 producing Foxp3^+^ Tregs, and largely prevented anaphylactic effects, including IgE production, mast cell activation and a drop in body temperature, both when applied prior to or after sensitization [166]. Liu and coworkers demonstrated that PLGA-based NPs decorated with mannose or an apolipoprotein (Apo) B-derived peptide after intravenous application into mice predominantly addressed LSECs via the mannose and the stabilin-2 receptor, respectively [165]. Both receptors are highly expressed by LSECs [14]. In allergy models, the prophylactic administration of NPs containing the model antigen ovalbumin (OVA) and coated with ApoB peptide resulted in the induction of OVA-specific Tregs that generated the anti-inflammatory cytokines, IL-10 and TGF-β [165]. In accordance, this vaccination approach prevented both anaphylactic and asthmatic reactions in the relevant models by preventing a Th2 response and concomitant IgE production. In a subsequent study the same LSEC-focused delivery approach, using the peanut allergen, Ara h2, as a cargo, yielded tolerogenic effects in a peanut anaphylaxis model [240].

In a follow-up study [32], the aforementioned LSEC-targeting tolerization approach using OVA as a model allergen showed similar efficacy as using PLGA-based NPs containing the mTOR inhibitory drug, rapamycin (immTOR), previously reported to passively target and tolerize DCs, thereby inducing Treg and attenuating B cell responses in mice [252] and cynomolgus monkeys [253]. However, a previous study demonstrated that immTOR NPs accumulated not only in the spleen, but also in the liver [252]. Ilyinskii et al. showed that immTOR addressed hepatocytes as well as all types of liver NPs [254]. Further, immTOR administration tolerized KCs, DCs and LSECs, as deduced from the downregulation of MHCII and costimulatory receptors, accompanied by the increased expression of the coinhibitory receptor, PD-L1. Further, hepatic T cell numbers were diminished, and the remaining population presented with elevated PD-1 levels. LSECs isolated from immTOR-injected mice inhibited T cell proliferation. In accordance with the broad anti-inflammatory effects of immTOR, pretreated mice displayed attenuated inflammation after the application of concavalin A.

In light of the general finding that many types of NPs accumulate in the liver [9], the study by Ilyinski and coworkers [254] underscores that nano-carriers previously shown to tolerize DCs in the spleen or lymph nodes [252,253] may also have a considerable impact on liver NPC populations. Further studies are required to revisit such nano-carriers in order to assess the potential synergistic effects of DCs and potentially tolerized liver NPCs, often not taken into account in previous ex vivo analyses, with regard to systemic tolerance induction.

## 3. Conclusions

The liver constitutes a highly suitable target organ for nanoformulations that enable drug delivery in a cell type-specific manner to reduce the adverse effects often observed in cases of systemic application. Based on detailed knowledge of cell type-specific surface receptors and intracellular signaling pathways targeting and reprogramming both hepatocytes for the treatment of metabolic diseases and liver NPCs to counteract fibrosis as a preliminary stage of cirrhosis/HCC, as well as of immunological traits, is now possible. Taking into account the extensive crosstalk between liver cell types, e.g., in HCC and TME combination therapies that co-target distinct cell types with appropriate nanodrugs and with clinically approved agents, such as immune checkpoint inhibitors, respectively, may achieve synergistic therapeutic effects. Moreover, besides the treatment of liver-restricted diseases the default protolerogenic state of liver NPCs, and here especially of LSECs, may be exploited to imprint systemic antigen-specific tolerance for the treatment of allergies and autoimmune diseases. Such approaches may be combined with nano-vaccines that target DCs in the periphery.

## Data Availability

Not applicable.

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
