# Peer review of "Liver Cell Type-Specific Targeting by Nanoformulations for Therapeutic Applications"

_ijms, 2023, doi:10.3390/ijms241411869_

Round 1

Reviewer 1 Report

The paper by Kaps and colleagues is a review discussing different strategies for liver disease targeting using nanomedicine. The paper is well written and the contents are clearly presented.

I have only two suggestions and comments:

1)      On page 5, two typos are present: “quiescent” and “ore-shell”. Please replace it with the correct words.

2)      On page 7, when listing the possible strategies for decorating NPs to target cancer cells, the Authors should also mention the possibility of functionalizing NPs with hyaluronic acid binding CD44, highly expressed on hepatic cancer cells. I suggest the Authors to cute the following papers:

-          Asai et al., CD44 standard isoform is involved in maintenance of cancer stem cells of a hepatocellular carcinoma cell line. Cancer Medicine 2019, DOI: 10.1002/cam4.1968.

-          Cannito et al., Hyaluronated and PEGylated Liposomes as a Potential Drug-Delivery Strategy to Specifically Target Liver Cancer and Inflammatory Cells. Molecules 2022, DOI: 10.3390/molecules27031062

-          Wang et al., CD44 antibody-targeted liposomal nanoparticles for molecular imaging and therapy of hepatocellular carcinoma. Biomaterials 2012,

https://doi.org/10.1016/j.biomaterials.2012.03.067

Author Response

Reviewer 1

The paper by Kaps and colleagues is a review discussing different strategies for liver disease targeting using nanomedicine. The paper is well written and the contents are clearly presented.

We thank the reviewer for his kind assessment of our manuscript.

I have only two suggestions and comments:

  • On page 5, two typos are present: “quiescent” and “ore-shell”. Please replace it with the correct words.

We thank the reviewer for these hints. We have thoroughly corrected the manuscript for typos.

2)      On page 7, when listing the possible strategies for decorating NPs to target cancer cells, the Authors should also mention the possibility of functionalizing NPs with hyaluronic acid binding CD44, highly expressed on hepatic cancer cells. I suggest the Authors to cute the following papers:

-          Asai et al., CD44 standard isoform is involved in maintenance of cancer stem cells of a hepatocellular carcinoma cell line. Cancer Medicine 2019, DOI: 10.1002/cam4.1968.

-          Cannito et al., Hyaluronated and PEGylated Liposomes as a Potential Drug-Delivery Strategy to Specifically Target Liver Cancer and Inflammatory Cells. Molecules 2022, DOI: 10.3390/molecules27031062

-          Wang et al., CD44 antibody-targeted liposomal nanoparticles for molecular imaging and therapy of hepatocellular carcinoma. Biomaterials 2012, https://doi.org/10.1016/j.biomaterials.2012.03.067

We have extented the according passage in the manuscript and have included table 1 that summarizes target receptors on liver tumor cells (including CD44) and liver NPC, respectively, which habe been successfully addressed by nanoformulations decorated with suitable ligands.

Reviewer 2 Report

Although the subject is relevant, I believe the authors did not explore it as its best. 
At first, regarding the manuscript title. The reader is invited to read about the state of art in immunotherapy. But in the manuscript this type of therapy, its advantages, challenges and exemplary sucessfully-used drugs are not mentioned. A review about what the term "immunotherapy" mean, what are the most common drugs, and what would be the challenges for its application is strongly recommended. 
Authors also mention, in Introduction that "This review aims to summarize strategies currently evaluated in preclinical in vivo models for the treatment of liver diseases with nanoformulations that may allow to overcome poor solubility of drugs by encapsulation and to codeliver distinct compounds to yield synergistic effects in a cell type-specific manner either via passive or active targeting by attachment of different receptor binding moieties on the NP surface", but it is not clear what are those strategies. It would be important to include a discussion about the receptors overexpressed in each cell type and how to target those receptors. For example, one of the most common approaches for HSC targeting: mannose-6-phosphate, that targets IGF/M6P receptor is not discussed. Other succesful liver cell-targeting approaches are not (or only slightly) mentioned, such as use of mannose for LSEC- and KC-targeting, or galactose/N-acetyl-galactosamine approaches for targeting hepatocytes

Authors also mention the co-delivery but the manuscript lacks of a discussion about what are the advantages and disadvantages or even the need of a combined-therapy instead of the administrations of these drugs separately.

I would also like to include some other suggestions for a future submission:
- In secton 2.1, the authors describe the evolution of liver damage to cirrhosis. The explanation is poor and lacks the description of the biochemical and cellular  mechanisms involved in this process. Additionally, it is not evidenced what would be the consequences of this process

- In section 2.1.1 authors mention that the most common route of infection for HCV and HBV is the exposure to contaminated blood. Although this affirmation is not wrong, it is important to clarify that HBV is mostly transmited through vertical contamination (mother to child during birth) and unprotected sexual intercourse while HCV is most commonly transmitted in adults, by sharing needles,  and blood transfusion

- In section 2.2, although there is no curative treatment for NASH, there are two FDA-approved drugs for treatment of NAFLD: pioglitazone and tocoferol. It would be interesting to discuss about their uses and limitations.

- Also, it whould have been interesting if the description of the nanoformulations was similar in each section. In some, authors mention the composition, while in others they only mention the type of nanostructure. In some, they include physico-chemical parameters (such as average diameter) while in other, this information is not presented.

All of it being said, I would suggest the authors to re-write the manuscript in a more comprehensive manner, and resubmit it.

There are some typos and some expressions are informal. I would suggest an English language review. 

Author Response

Although the subject is relevant, I believe the authors did not explore it as its best.

At first, regarding the manuscript title. The reader is invited to read about the state of art in immunotherapy. But in the manuscript this type of therapy, its advantages, challenges and exemplary sucessfully-used drugs are not mentioned. A review about what the term "immunotherapy" mean, what are the most common drugs, and what would be the challenges for its application is strongly recommended.

We thank the reviewer for his suggestion. We have altered the title of the manuscript (“nanoformulations for immunotherapy” => “… for therapeutic applications”) to reflect the broader subject of the review.

Authors also mention, in Introduction that "This review aims to summarize strategies currently evaluated in preclinical in vivo models for the treatment of liver diseases with nanoformulations that may allow to overcome poor solubility of drugs by encapsulation and to codeliver distinct compounds to yield synergistic effects in a cell type-specific manner either via passive or active targeting by attachment of different receptor binding moieties on the NP surface", but it is not clear what are those strategies. It would be important to include a discussion about the receptors overexpressed in each cell type and how to target those receptors. For example, one of the most common approaches for HSC targeting: mannose-6-phosphate, that targets IGF/M6P receptor is not discussed. Other succesful liver cell-targeting approaches are not (or only slightly) mentioned, such as use of mannose for LSEC- and KC-targeting, or galactose/N-acetyl-galactosamine approaches for targeting hepatocytes.

We thank the reviewer for this important hint. We have included table 1 to summarize which receptors on HCC and liver NPC have been addressed so far by targeting nanoformulations. This talbe comprises the receptors/targeting ligands mentioned by the reviewer.

Authors also mention the co-delivery but the manuscript lacks of a discussion about what are the advantages and disadvantages or even the need of a combined-therapy instead of the administrations of these drugs separately.

We have extended the according text passage in the introduction.

I would also like to include some other suggestions for a future submission:

- In secton 2.1, the authors describe the evolution of liver damage to cirrhosis. The explanation is poor and lacks the description of the biochemical and cellular  mechanisms involved in this process. Additionally, it is not evidenced what would be the consequences of this process.

Section 2.1 briefly summarizes the different etiologies of viral hepatitis. We have renamed the title to viral hepatitis as this section sets a focus on viral hepatitis and does not extensively discuss other etiologies. We agree that cellular mechanisms of fibrosis are not discussed in detail but this is somehow beyond of the intended scope of the section. However, we have added a recent reference, which discusses nicely these mechanisms in detail (F Tacke et al. 2023). The clinically most relevant consequence of cirrhosis is portal hypertension and we have added this information to the section.

- In section 2.1.1 authors mention that the most common route of infection for HCV and HBV is the exposure to contaminated blood. Although this affirmation is not wrong, it is important to clarify that HBV is mostly transmited through vertical contamination (mother to child during birth) and unprotected sexual intercourse while HCV is most commonly transmitted in adults, by sharing needles,  and blood transfusion.

We agree and have changed the passage accordingly.

- In section 2.2, although there is no curative treatment for NASH, there are two FDA-approved drugs for treatment of NAFLD: pioglitazone and tocoferol. It would be interesting to discuss about their uses and limitations.

We could find no reference that the drugs have been approved for patients with NAFLD by the FDA. We have mention that both drugs showed an effect for hepatic steatosis. However, the discussion of the drug’s limitations is beyond of the scope of our review. We want to focus on nanoparticular therapies in NAFLD/NASH and just want to give a basic introduction of fatty liver disease.

- Also, it would have been interesting if the description of the nanoformulations was similar in each section. In some, authors mention the composition, while in others they only mention the type of nanostructure. In some, they include physico-chemical parameters (such as average diameter) while in other, this information is not presented.

We provide now only the general type of nanocarrier without going into too much detail.

All of it being said, I would suggest the authors to re-write the manuscript in a more comprehensive manner, and resubmit it. Comments on the Quality of English Language: There are some typos and some expressions are informal. I would suggest an English language review.

We have thoroughly revised the manuscript to eliminate typos and to avoid unusual expressions.